# Impact of the COVID-19 Pandemic on Influenza Vaccination and Associated Factors among Pregnant Women: A Cross-Sectional Study in Korea

**DOI:** 10.3390/vaccines11030512

**Published:** 2023-02-22

**Authors:** Boyeon Kim, Eunyoung Kim

**Affiliations:** Data Science, Evidence-Based and Clinical Research Laboratory, Department of Health, Social and Clinical Pharmacy, College of Pharmacy, Chung-Ang University, Seoul 06974, Republic of Korea

**Keywords:** influenza, vaccination, pregnant women, COVID-19, knowledge, acceptance

## Abstract

Pregnant women are vulnerable to developing influenza complications. Influenza vaccination during pregnancy is crucial to avoid infection. The COVID-19 pandemic might exacerbate fear and anxiety in pregnant women. The purpose of this study was to evaluate the effect of the COVID-19 pandemic on influenza vaccination and determine the factors associated with influenza vaccine acceptance among pregnant women in Korea. We conducted a cross-sectional study using an online survey in Korea. A survey questionnaire was distributed among pregnant or postpartum women within 1 year after delivery. Multivariate logistic regression analysis was performed to identify the factors associated with influenza vaccination among pregnant women. A total of 351 women were included in this study. Of them, 51.0% and 20.2% were vaccinated against influenza and COVID-19 during pregnancy, respectively. The majority of participants who had a history of influenza vaccination reported that the COVID-19 pandemic did not affect (52.3%, *n* = 171) or increased the importance (38.5%, *n* = 126) of their acceptance of the influenza vaccine. Factors associated with influenza vaccine acceptance were knowledge of influenza vaccine (OR 1.21; 95% CI 1.09, 1.35), trust in healthcare providers (OR 2.57; 95% CI 1.43, 4.65), and COVID-19 vaccination during pregnancy (OR 6.11, 95% CI 2.86, 13.01). Participants were more likely to accept the influenza vaccine when they received a COVID-19 vaccine during pregnancy, but the rate of influenza vaccination was not affected by the COVID-19 pandemic. This study showed that the COVID-19 pandemic did not influence influenza vaccine uptake in the majority of pregnant women in Korea. The results emphasize the necessity of appropriate education for pregnant women to enhance awareness of vaccination.

## 1. Introduction

Influenza can develop into severe illness and the influenza-associated all-cause deaths from 2009 to 2016 were found to be 10.59 per 100,000 people annually in Korea [1]. Pregnant women are at high risk for developing complications from influenza, including hospitalization, intensive care unit admission, and death [2]. Infants of pregnant women with influenza infection are more likely to be born preterm, which is associated with neonatal morbidity and mortality [3,4]. Thus, pregnant women should receive the influenza vaccination as a critical strategy to prevent infection; their uptake of the influenza vaccine should also be increased [5]. Additionally, receiving the influenza vaccine during pregnancy effectively prevents influenza infection in infants up to 6 months of age who are ineligible for influenza vaccination [6,7]. Therefore, it is recommended that pregnant women receive an influenza vaccination [8,9].

The number of confirmed cases and deaths due to the COVID-19 pandemic is still increasing [10]. In Korea, 35,086 reported COVID-19 cases and 112 reported hospitalizations on 26 January 2023 [11]. As of June 2022, 211,551 pregnant women had contracted COVID-19 in the United States, contributing to a total of 33,123 hospitalizations and 295 deaths [12]. Studies have suggested that pregnant women with COVID-19 are at a higher risk for hospitalization, mechanical ventilation, intensive care, and death compared to non-pregnant women [13,14,15]. This can negatively impact pregnant women, causing fear and anxiety [16]. In addition, the medical burden has increased due to the COVID-19 pandemic, causing delayed access to healthcare services [17], which is clearly associated with adverse outcomes among pregnant women [18]. Although pregnant women are classified as a COVID-19-vulnerable population, original clinical trials evaluating the efficacy and safety of the COVID-19 vaccine excluded pregnant and lactating women [19,20,21].

Emerging data have provided no evidence of increased adverse outcomes—including miscarriage, preterm birth, or neonatal intensive care unit admission—among pregnant women who received the COVID-19 vaccine [22,23,24]. COVID-19 vaccination during pregnancy reduces the risks of infection and COVID-19-related hospitalization [25,26,27]. Furthermore, completing a two-dose COVID-19 vaccine regimen during pregnancy can help prevent hospitalization in infants up to 6 months of age [28]. Currently, the COVID-19 vaccination is recommended for pregnant women [29,30,31].

Despite recommendations for vaccination among pregnant women, the acceptance of both influenza and COVID-19 vaccines is low [32,33,34,35,36,37,38,39,40,41]. Previous studies have investigated the factors influencing influenza or COVID-19 vaccine acceptance among pregnant women [35,36,42,43,44,45,46,47,48]. A few studies have examined the knowledge and attitudes towards influenza vaccination among pregnant women during the COVID-19 pandemic [49,50]. However, no studies have investigated the acceptance of the influenza vaccine during the COVID-19 pandemic among pregnant women in Korea.

This study aimed to evaluate the acceptance of the influenza vaccine during the COVID-19 pandemic and determine the factors associated with the influenza vaccine acceptance among pregnant women in Korea. This study also examined the knowledge of the influenza vaccine, trust in healthcare providers, and health beliefs regarding the influenza vaccine among pregnant women during the COVID-19 pandemic.

## 2. Materials and Methods

### 2.1. Study Design and Participants

This cross-sectional study was conducted using an online survey in Korea between 1 April 2022 and 15 April 2022. A self-administered questionnaire was created online and distributed through the largest online panel operated by Tillion Pro in Korea [51]. Pregnant or postpartum women within one year of delivery were eligible and participation in the survey was voluntary. To obtain a representative sample of the Korean population, participants were recruited via stratification of regional residences. Individuals under 12 weeks of gestation and who were giving birth to their first child were excluded from the analysis. The sample size was calculated with the following assumptions: the acceptance rate of the influenza vaccine during pregnancy was 62.3% [46], with a confidence interval of 95% and an alpha of 0.05. The sample size was 361. The participants’ identifying information was removed from the data analyzed in this study to ensure confidentiality and anonymity. The study was approved by the Chung-Ang University Institutional Review Board (1041078-202203-HR-063).

### 2.2. Survey Questionnaires

A structured questionnaire developed based on a literature review was distributed to participants covering the following information: background characteristics, characteristics related to vaccine acceptance, knowledge of the influenza vaccine, trust in healthcare providers, and health beliefs related to the influenza vaccine. A pilot test among a sample of 50 pregnant women was conducted as non-facial validation to enhance the clarity and validity of the survey questions.

#### 2.2.1. Background Characteristics

Background characteristics were divided into sociodemographic and pregnancy characteristics. Sociodemographic characteristics consisted of age, region of residence, religious affiliation, education, current employment status, economic status, and self-rated health status. Pregnancy characteristics included gestational age, parity, history of miscarriage, complications during pregnancy, and respiratory disease status.

#### 2.2.2. Characteristics Related to Vaccine Acceptance

The participants were asked whether they were vaccinated for influenza and/or COVID-19 during pregnancy. The acceptance of influenza vaccine among pregnant women was collected by asking the question, ‘Have you been vaccinated against influenza during pregnancy?’ We used the question ‘Have you ever been vaccinated against influenza?’ to collect the participants’ histories of influenza vaccination. Their reasons for deciding not to receive a vaccination during pregnancy were also collected using a multiple-choice format in the questionnaire. Side effects after vaccination were also surveyed.

#### 2.2.3. Knowledge Regarding Influenza Vaccine

Eleven items were used to examine participants’ knowledge of the influenza vaccine, including the effectiveness and safety of the influenza vaccine during pregnancy, the number of influenza vaccinations, and the recommendations for influenza vaccination during pregnancy. Three responses could be chosen: yes, no, or not sure. If the correct answer was chosen, one point was added to the total score. The total score ranged from 0 to 11.

#### 2.2.4. Trust in Healthcare Providers

Trust in healthcare providers of pregnant women was collected by providing the following two items: (1) If healthcare providers recommend the influenza vaccination, I would get vaccinated; (2) I can trust healthcare providers to give appropriate and effective treatment that is best for me. Answers were scored on a 4-point scale ranging from 1 = do not agree at all to 4 = strongly agree.

#### 2.2.5. Health Beliefs Related to Influenza and Vaccination

Four categories were included: (1) perceived susceptibility to influenza infection; (2) perceived severity of influenza infection; (3) perceived benefits of influenza vaccination; and (4) perceived barriers to influenza vaccination. The participants were asked to rate each item on a 4-point scale ranging from 1 = do not agree at all to 4 = strongly agree.

### 2.3. Statistical Analysis

Data were statistically analyzed with SPSS version 27.0. Descriptive statistics were presented as frequencies and percentages for categorical variables and means ± standard deviations for continuous variables. Cronbach’s α coefficients were calculated to assess the internal consistency and reliability of the questionnaire for knowledge, trust in healthcare providers, and health beliefs. Chi-square or t-tests were performed to examine the significance of the differences between the background characteristics of the participants and their influenza vaccine acceptance. A *t*-test was conducted to compare study variables, including knowledge, trust in healthcare providers, and health beliefs related to the acceptance of the influenza vaccine. Pearson’s correlation coefficients and Point-biserial correlation coefficients were calculated to evaluate significant correlations between study variables. Multivariate logistic regression analysis was performed to identify the factors associated with the acceptance of the influenza vaccine among pregnant women during the COVID-19 pandemic. Variables without multicollinearity in univariate logistic regression analyses were entered in multivariate logistic regression analyses. Background characteristics were included in Model 1, and study variables were entered in Model 2. Odds ratio (OR) and 95% confidence interval (CI) were calculated and unstandardized regression coefficients (B) were presented in multivariate logistic regression analysis. Two-tailed *p* < 0.05 was used to define statistical significance.

## 3. Results

### 3.1. Characteristics of the Study Population

A total of 351 women who were either pregnant (*n* = 201, 57.3%) or postpartum (*n* = 150, 42.7%) were included in this study (Table 1). One hundred percent of the study participants (*n* = 351) responded to all of the questions. Of these women, 50.4% were less than 35 years old, and 49.3% lived in a metropolitan area. The mean gestation age among the participating pregnant women was 23.88 weeks. The majority (80.4%) of women had experienced at least one delivery, and 27.1% had a history of miscarriage. Of the total participants, 51.0% were vaccinated against influenza during pregnancy. Most women had a history of influenza vaccination (*n* = 327, 93.2%). The COVID-19 vaccine acceptance rate during pregnancy was only 20.2% among participants.

### 3.2. Effect of the COVID-19 Pandemic on Influenza Vaccination

In this study, 52.3% (*n* = 171/327) of the participants who had a history of influenza vaccination reported that the COVID-19 pandemic did not affect their acceptance of the influenza vaccine. In contrast, 38.5% (*n* = 126/327) of them indicated that the COVID-19 pandemic had increased the importance of their acceptance of the influenza vaccine. Only 9.2% (*n* = 30/327) of them did not believe that influenza vaccination would be necessary.

The majority of women who had never been vaccinated for influenza (91.7%, *n* = 22/24) reported that the COVID-19 pandemic did not affect their acceptance of the influenza vaccine.

### 3.3. Acceptance of the Influenza Vaccine by Sociodemographic and Pregnancy Characteristics

The mean age of the participants who were vaccinated against influenza was significantly higher than the mean age of the unvaccinated participants (*p* = 0.007). Furthermore, 36.9% of women vaccinated against influenza had experienced two or more births. Pregnant women with a history of miscarriage, history of education regarding vaccination, willingness to receive the influenza vaccine during pregnancy, history of influenza vaccination, and vaccination against COVID-19 during pregnancy were more likely to accept the influenza vaccination (*p* < 0.05, Table 2). Cronbach’s α coefficients were in the range of 0.706–0.91.

### 3.4. Acceptance of the Influenza Vaccine by Knowledge, Trust in Healthcare Providers, and Health Beliefs

Pregnant women who were vaccinated against influenza scored significantly higher than unvaccinated pregnant women on knowledge regarding the influenza vaccine, trust in healthcare providers, perceived susceptibility to influenza infection, perceived severity of influenza infection, and perceived benefits of influenza vaccination (*p* < 0.05, Table 3).

### 3.5. Correlation between Knowledge, Trust in Healthcare Providers, Health Beliefs and Acceptance of the Influenza Vaccine

Significant correlations were observed between study variables (Table 4). Influenza vaccination was significantly related to increased knowledge regarding the influenza vaccine, trust in healthcare providers, perceived susceptibility, perceived severity, and perceived benefits (*p* < 0.01). Knowledge regarding the influenza vaccine and perceived susceptibility was positively related to trust in healthcare providers. Perceived severity was positively related not only to trust in healthcare providers but also to perceived susceptibility. Perceived benefits were positively related to knowledge regarding the influenza vaccine, trust in healthcare providers, perceived susceptibility, and perceived severity. Perceived barriers were negatively related to knowledge regarding the influenza vaccine, while perceived barriers had a significantly positive correlation with perceived susceptibility and perceived severity.

### 3.6. Factors Associated with Acceptance of the Influenza Vaccine during Pregnancy

Age, parity, history of miscarriage (1 = yes; 0 = no), history of education regarding vaccination (1 = yes; 0 = no), and acceptance of the COVID-19 vaccine during pregnancy (1 = yes; 0 = no) were included in Model 1. Knowledge regarding the influenza vaccine, trust in healthcare providers, perceived susceptibility, perceived severity, and perceived benefits were entered in Model 2 (Figure 1).

Regression Model 1 fitted the data well (χ^2^(5) = 10.77; *p* > 0.05), explaining 27.6% of the variance in the acceptance of the influenza vaccine during pregnancy (Nagelkerke’s R^2^ = 0.276). In Model 1, age (OR 1.07; 95% CI 1.00, 1.14), parity (OR 2.27; 95% CI 1.60, 3.20), history of miscarriage (OR 1.74; 95% CI 1.00, 3.00), history of education regarding the influenza vaccine (OR 1.75; 95% CI 1.06, 2.89), and acceptance of the COVID-19 vaccine during pregnancy (OR 6.07; 95% CI 3.04, 12.13) significantly increased the chance for acceptance of the influenza vaccine during pregnancy.

Regression Model 2 also fitted the data well (χ^2^(10) = 8.26; *p* > 0.05), explaining 40.6% of the variance in the acceptance of the influenza vaccine during pregnancy (Nagelkerke’s R^2^ = 0.406). In Model 2, parity (OR 2.22; 95% CI 1.53, 3.21), history of miscarriage (OR 2.29; 95% CI 1.26, 4.18), history of education regarding the influenza vaccine (OR 1.85; 95% CI 1.07, 3.18), acceptance of the COVID-19 vaccine during pregnancy (OR 6.11; 95% CI 2.86, 13.01), knowledge regarding the influenza vaccine (OR 1.21; 95% CI 1.09, 1.35), and trust in healthcare providers (OR 2.57; 95% CI 1.43, 4.65) significantly increased the chance for acceptance of the influenza vaccine during pregnancy. However, health beliefs including perceived susceptibility, perceived severity, and perceived benefits were not significantly associated with influenza vaccine acceptance.

## 4. Discussion

To our knowledge, this study is the first to assess the impact of the COVID-19 pandemic on influenza vaccine uptake among pregnant women in Korea. This study also investigated the factors associated with the acceptance of the influenza vaccine, including knowledge of the influenza vaccine, trust in healthcare providers, and COVID-19 vaccine uptake.

The acceptance rate of the influenza vaccine was 51% among pregnant women during the COVID-19 pandemic. Before this pandemic in 2018–2019, the influenza vaccination rates in Korea were 59.3–62.3%, which were similar to the results in this study [46,52]. This result could be attributed to the report of the majority of participants who indicated that the COVID-19 pandemic did not affect their acceptance of the influenza vaccine in this study. Conversely, in a recent qualitative interview study that explored the impact of the COVID-19 pandemic on pregnant women’s attitudes toward maternal vaccines in the United Kingdom, participants felt that the COVID-19 pandemic has increased the importance of maternal vaccination, including the influenza vaccine [53]. Another study showed an increase in the acceptance rate of the influenza vaccine during the COVID-19 pandemic among pregnant women [54]. Therefore, it is possible that pregnant women in Korea lacked sufficient education regarding influenza vaccination during the pandemic.

In this study, only 20.2% of women were vaccinated against COVID-19 during pregnancy in Korea. These findings were similar to previous results of studies on pregnant women in other countries [35,36,55,56]. However, a recent study conducted in 16 countries showed that the acceptance rate of the COVID-19 vaccine among pregnant women was 52% (range: 28.8–84.4%) [57]. Another study in China reported a higher rate of COVID-19 vaccination among pregnant women (77.4%) [58]. Therefore, the acceptance rate of the COVID-19 vaccine substantially varied globally.

The COVID-19 vaccination rate was low among pregnant women in Korea likely because pregnant women have concerns about the effectiveness and safety of the COVID-19 vaccine [35,36,47,55,56]. In the present study, more than 60% (*n* = 172) of the participants who did not receive the COVID-19 vaccine during pregnancy had concerns regarding the lack of data about vaccine efficacy or its potential side effects, and 27.5% (*n* = 77) feared that it would harm the fetus.

Our study showed that the vaccinated population was slightly older than the unvaccinated population (34.87 and 33.76 years old, respectively; *p* = 0.007). The acceptance rate of the influenza vaccine among pregnant women younger than 30 years was 41.3%; whereas, for women older than 40 years, it was 64.3%. In France, a previous study showed that older pregnant women are more likely to be vaccinated against influenza [34]. Our study also found that the participants with high parity or a history of miscarriage were more likely to accept the influenza vaccine. Conversely, influenza vaccination is not influenced by parity and history of miscarriage in several studies [34,46,59]. In the present study, more than two-thirds of women who had a history of education regarding vaccination were vaccinated against influenza. This finding was consistent with the result of a previous study conducted in Korea [46]. Therefore, a recommendation that emerges from this study is that accurate information about vaccination through education should be provided to promote vaccine uptake. Additionally, pregnant women vaccinated against COVID-19 were more likely to accept the influenza vaccination. Similar results were reported in a previous study in Poland [50]. These findings suggested a possible association between receiving influenza and COVID-19 vaccinations among pregnant women.

This study also found that pregnant women vaccinated against influenza had a higher knowledge score on the influenza vaccine than unvaccinated women. Similar results were observed in previous studies. One study showed that pregnant women with a higher knowledge score regarding the influenza vaccine and influenza infection were more likely to accept the influenza vaccination [59]. Another study reported that improving pregnant women’s knowledge about vaccination has helped to increase the acceptance rate of the influenza vaccine [60]. These findings are attributed to the positive association of knowledge about influenza vaccination with acceptance rate [61,62,63]. Healthcare providers play a key role in helping women make an informed choice about vaccines when they are deciding whether to accept the influenza vaccine during pregnancy [64]. In the present study, trust in healthcare providers was significantly higher in pregnant women vaccinated against influenza than in unvaccinated pregnant women. Pregnant women were more likely to get vaccinated when they had a higher perceived susceptibility and severity of infection regarding influenza and know of more benefits from the influenza vaccination. These results are in line with previous studies [45,49,59]. Therefore, pregnant women should receive information about susceptibility to influenza infection and the benefits of influenza vaccination. Moreover, they should recognize the severity of influenza infection.

This study confirmed that the acceptance of the influenza vaccine during pregnancy is associated with various factors, including knowledge of the influenza vaccine and trust in healthcare providers. Pregnant women who received education regarding the influenza vaccine and had a high level of knowledge about the influenza vaccine were more likely to get vaccinated against influenza. These findings were similar to previous studies conducted among pregnant women [59,65,66]. In the present study, trust in healthcare providers was also significantly correlated with the acceptance of the influenza vaccine among pregnant women. A previous qualitative evidence synthesis showed that pregnant women’s trust in healthcare providers can promote making the decision to get vaccinated against influenza [67]. Therefore, acceptance of the influenza vaccine could be improved by increasing the opportunity for healthcare providers who established a good relationship of trust with pregnant women to provide accurate information regarding the influenza vaccine. Furthermore, evidence-based and professional training or education is a necessity for healthcare providers to promote the discussion of influenza vaccination and address the concerns of pregnant women [67].

Participants were six times more likely to accept the influenza vaccine when they received a COVID-19 vaccine during pregnancy. This study suggested that influenza vaccination was strongly associated with COVID-19 vaccination. On the other hand, a recent study performed on pregnant women in the United States reported that having had or intending to have an influenza vaccine is highly negatively associated with COVID-19 vaccine hesitancy [68].

This study has several strengths. This study is the first to explore the impact of the COVID-19 pandemic on pregnant women’s influenza vaccine uptake in Korea. This study also determined the factors associated with influenza vaccine acceptance among pregnant women in Korea during the COVID-19 pandemic. Additionally, in contrast to previous studies performed in healthcare settings on influenza vaccine acceptance, this is the first survey that uses the online panel stratified by region in Korea. However, this study has several limitations. First, all data were self-reported and could not be validated independently, resulting in a potential reporting bias in the results. Second, although the sample population was stratified by region of residence to minimize potential selection bias, the study population might not be representative of all pregnant women in Korea. The participants were not randomly selected because a panel survey platform was used to perform the online survey. Lastly, the cross-sectional design of this study could not establish causal relationships between factors.

## 5. Conclusions

This cross-sectional study showed that the influenza vaccine uptake of pregnant women was not influenced by the COVID-19 pandemic in Korea. Even though pregnant women are vulnerable to both influenza and COVID-19, we found that the acceptance rates of these vaccines were still low in Korea. These results suggest that education for pregnant women was not substantial enough to encourage vaccination, despite the COVID-19 pandemic. In addition, knowledge of the influenza vaccine, trust in healthcare providers, and receiving the COVID-19 vaccination were significantly associated with the acceptance of the influenza vaccine among pregnant women. Therefore, additional educational campaigns are needed to enhance awareness of the importance of vaccination and provide information about the vaccine by reliable healthcare providers. This will improve vaccine uptake in pregnant women in Korea.

## Figures and Tables

**Figure 1 vaccines-11-00512-f001:**
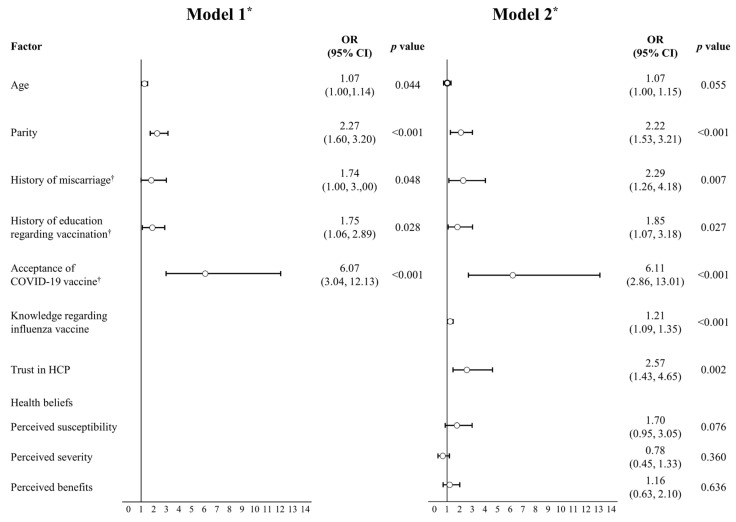
Factors associated with the acceptance of the influenza vaccine during pregnancy. *n* = 351. * Model 1: Hosmer & Lemeshow χ^2^(5) = 10.77, *p* > 0.05, Nagelkerke’s R^2^ = 0.276; Model 2: Hosmer & Lemeshow χ^2^(10) = 8.26, *p* > 0.05, Nagelkerke’s R^2^ = 0.406. ^†^ yes = 1; no = 0. OR: odds ratio; HCP: healthcare providers.

**Table 1 vaccines-11-00512-t001:** Sociodemographic and pregnancy characteristics of the study population.

Characteristics	N (%)
**Total**	351 (100)
** *Sociodemographic characteristics* **	
**Age (years)**	
<30	46 (13.1)
30–34	131 (37.3)
35–39	146 (41.6)
40–44	28 (8.0)
Mean ± SD	
**Region**	
Metropolitan area	173 (49.3)
Non-metropolitan area	178 (50.7)
**Religious affiliation**	
Yes	165 (47.0)
No	186 (53.0)
**Education**	
High school or below	20 (5.7)
Bachelor’s degree	282 (80.3)
Postgraduate degree	49 (14.0)
**Employed**	
Yes	225 (64.1)
No	126 (35.9)
**Economic status (per month)**	
<3,000,000 won	54 (15.4)
3,000,000–4,000,000 won	67 (19.1)
4,000,000–5,000,000 won	72 (20.5)
5,000,000–6,000,000 won	64 (18.2)
≥6,000,000 won	94 (26.8)
**Self-rated health status**	
Poor	11 (3.1)
Fair	243 (69.2)
Good	97 (27.6)
** *Pregnancy characteristics* **	
**Current pregnancy**	
Yes	201 (57.3)
No	150 (42.7)
**Gestational age**	
First trimester (1–12 weeks)	14 (7.0)
Second trimester (13–28 weeks)	122 (60.7)
Third trimester (29–40 weeks)	65 (32.3)
**Parity**	
0	69 (19.7)
1	188 (53.6)
≥2	94 (26.8)
**History of miscarriage**	
Yes	95 (27.1)
No	256 (72.9)
**Complication during pregnancy**	
Yes	15 (4.3)
No	336 (95.7)
**Respiratory disease status**	
Yes	7 (2.0)
No	344 (98.0)
** *Characteristics related to vaccine acceptance* **	
**History of education regarding vaccination**	
Yes	127 (36.2)
No	224 (63.8)
**Willingness to receive influenza vaccine** **during pregnancy**	
Yes	228 (65.0)
No	123 (35.0)
**Number of influenza vaccination in the past**	
1	101 (28.8)
2	128 (36.5)
≥3	98 (27.9)
Never	24 (6.8)
**Acceptance of COVID-19 vaccine during pregnancy**	
Yes	71 (20.2)
No	280 (79.8)

**Table 2 vaccines-11-00512-t002:** Acceptance of the influenza vaccine among pregnant women by sociodemographic and pregnancy characteristics.

	Acceptance of the Influenza Vaccine during Pregnancy (%)		
Characteristics	Yes	No	*p*-Value
**Total**	179 (51.0)	172 (49.0)	
** *Sociodemographic characteristics* **			
**Age (years)**			
<30	19 (10.6)	25 (15.7)	0.083
30–34	60 (33.5)	71 (41.3)	
35–39	82 (45.8)	64 (37.2)	
40–44	18 (10.1)	10 (5.8)	
Mean ± SD	34.87 ± 3.82	33.76 ± 3.84	0.007
**Region**			
Metropolitan area	88 (49.2)	85 (49.4)	0.962
Non-metropolitan area	91 (50.8)	87 (50.6)	
**Religious affiliation**			
Yes	88 (49.2)	77 (44.8)	0.410
No	91 (50.8)	95 (55.2)	
**Education**			
High school or below	11 (6.1)	9 (5.2)	0.584
Bachelor’s degree	140 (78.2)	142 (82.6)	
Postgraduate degree	28 (15.6)	21 (12.2)	
**Employed**			
Yes	116 (64.8)	109 (63.4)	0.780
No	63 (35.2)	63 (36.6)	
**Economic status (per month)**			
<3,000,000 won	29 (16.2)	25 (14.5)	0.693
3,000,000–4,000,000 won	33 (18.4)	34 (19.8)	
4,000,000–5,000,000 won	40 (22.3)	32 (18.6)	
5,000,000–6,000,000 won	28 (15.6)	36 (20.9)	
≥6,000,000 won	49 (27.4)	45 (26.2)	
**Self-rated health status**			
Poor	6 (3.4)	5 (2.9)	0.828
Fair	126 (70.4)	117 (68.0)	
Good	47 (26.3)	50 (29.1)	
** *Pregnancy characteristics* **			
**Current pregnancy**			
Yes	96 (53.6)	105 (61.0)	0.160
No	83 (46.4)	67 (39.0)	
**Gestational age**			
First trimester (1–12 weeks)	7 (7.3)	7 (6.7)	0.451
Second trimester (13–28 weeks)	54 (56.3)	68 (64.8)	
Third trimester (29–40 weeks)	35 (36.5)	30 (28.6)	
**Parity**			
0	20 (11.2)	49 (28.5)	<0.001
1	93 (52.0)	95 (55.2)	
≥2	66 (36.9)	28 (16.3)	
**History of miscarriage**			
Yes	59 (33.0)	36 (20.9)	0.011
No	120 (67.0)	136 (79.1)	
**Complication during pregnancy**			
Yes	9 (5.0)	6 (3.5)	0.476
No	170 (95.0)	166 (96.5)	
**Respiratory disease status**			
Yes	5 (2.8)	2 (1.2)	0.275
No	174 (97.2)	170 (98.8)	
** *Characteristics related to vaccine acceptance* **			
**History of education regarding vaccination**			
Yes	81 (45.3)	46 (26.7)	<0.001
No	98 (54.7)	126 (73.3)	
**Willingness to receive influenza vaccine** **during pregnancy**			
Yes	164 (91.6)	64 (37.2)	<0.001
No	15 (8.4)	108 (62.8)	
**Number of influenza vaccination in the past**			
1	60 (33.5)	41 (23.8)	<0.001
2	67 (37.4)	61 (35.5)	
≥3	52 (29.1)	46 (26.7)	
Never	-	24 (14.0)	
**Acceptance of COVID-19 vaccine during pregnancy**			
Yes	58 (32.4)	13 (7.6)	<0.001
No	121 (67.6)	159 (92.4)	

**Table 3 vaccines-11-00512-t003:** Acceptance of the influenza vaccine by knowledge, trust in healthcare providers, and health beliefs.

		Acceptance of the Influenza Vaccine during Pregnancy (%)		
Variable		Yes	No	*p*-Value
**Knowledge regarding influenza vaccine**		0.69 ± 0.21	0.56 ± 0.23	<0.001
**Trust in HCP**		3.03 ± 0.55	2.68 ± 0.59	<0.001
**Health beliefs**	Perceived susceptibility	2.56 ± 0.58	2.36 ± 0.56	0.001
	Perceived severity	2.59 ± 0.65	2.41 ± 0.61	0.01
	Perceived benefits	2.96 ± 0.51	2.70 ± 0.59	<0.001
	Perceived barriers	2.48 ± 0.54	2.46 ± 0.53	0.715

HCP: healthcare providers.

**Table 4 vaccines-11-00512-t004:** Correlation between knowledge, trust in healthcare providers, health beliefs, and acceptance of the influenza vaccine.

Variable		1	2	3	4	5	6	7
**1**	**Knowledge**	1						
**2**	**Trust in HCP**	0.291 *	1					
**3**	**Susceptibility**	−0.016	0.272 *	1				
**4**	**Severity**	0.038	0.329 *	0.626 *	1			
**5**	**Benefits**	0.262 *	0.586 *	0.349 *	0.340 *	1		
**6**	**Barriers**	−0.267 *	−0.030	0.361 *	0.404 *	−0.059	1	
7	**Influenza vaccination** ^†^	0.264 *	0.296 *	0.173 *	0.138 *	0.231 *	0.020	1

*n* = 351. * *p* < 0.01. ^†^ yes = 1; no = 0. The numbers in the first row correspond to the numbers of variables in the first column. HCP: healthcare providers.

## Data Availability

All data presented in the study are available from the corresponding author upon request.

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
