# Peer review of "Impact of the COVID-19 Pandemic on Influenza Vaccination and Associated Factors among Pregnant Women: A Cross-Sectional Study in Korea"

_vaccines, 2023, doi:10.3390/vaccines11030512_

Round 1

Reviewer 1 Report

Review of Manuscript ID: vaccines-2163725

The manuscript is well written giving appropriate and recent references where needed. Few comments are given below that require attention:

1.     The Abstract states nationwide survey, however, only 351 women were included in the online survey. A meagre number of subjects, if it is a nationwide survey. A sample size calculation should be given, and the word “nationwide’ should be deleted. Or a justification could be provided for the small sample size in a nationwide online survey.

2.     Introduction: The following sentence needs to be rephrased keeping in sync with the objective of the study. ” no studies have investigated the association between the acceptance of influenza and COVID-19 vaccines among pregnant women.” Replaced with “..acceptance of influenza vaccine during the COVID-19 pandemic”.

3.     Materials and Methods: Under 2.2.3. If the correct answer was chosen, participants were given one point” how would the authors know/verify whether the participant gave correct answer? Please clarify the language in the MS.

4.     RESULTS:

-It is not clear what is meant by “response rate is 100%”. All participants (n=351) who have enrolled in the study, 100% of them have responded to all questions.

-Please clarify the two sentences: (L 133-135) Of the total participants, 51.0% were vaccinated against influenza during pregnancy. Most women had a history of influenza vaccination (n=327, 93.2%). Does this mean that these women had earlier received influenza vaccine prior to the present/latest pregnancy? If so, was the earlier influenza vaccination taken during earlier pregnancy or non-pregnant state?

Table 1: what is meant by Religion, yes or no. Either include the various religions (Buddhist, Christian etc) or write Religious minded, Yes/No.

-Line 155: vaccination against COVID-19 ; does this mean ‘history of vaccination against COVID-19’ ? or “acceptance of COVID-19 vaccine during pregnancy”?

- The term ‘perceived severity’ is actually not clear. Does that mean perceived severity of side effects of vaccination or perceived severity of infection? Please define and elaborate these phrases in the Method section. (a) perceived severity (of influenza infection?); (b) perceived benefits (of influenza vaccination?); (c) perceived susceptibility (to influenza infection?); (d) perceived barriers (to vaccination?)

5.     DISCUSSION:

When discussing about trust on healthcare providers, who can enhance influenza vaccine uptake by pregnant women, emphasis on adequate and high-quality training of healthcare providers (especially at rural levels, and at lower stage of education, e.g. health care workers) should be made. It appears from various publications including this manuscript that healthcare providers play a big role in building trust on the health care system and use of the provided services by the public.

- Showing ‘strong association between pregnant women’s influenza vaccination and  COVID-19 vaccination’, cannot be a strength of the study. This is simply a finding of the study. Similarly, the first study to explore association in Korea can also not be a strength. The authors need to find out strength of the study and then state.

Minor comments

Line 131-132: add ‘age’ or ‘period’ after “mean gestation”.

Line 133: majority (80.4%) of women had experienced at least one birth”; here ‘birth’ can be replaced by ‘delivery’.

Line 202-203: increased the odds for influenza vaccination during pregnancy; can be added as increased the chance for acceptance of (or willingness to take) influenza vaccination during pregnancy.

Line 210-211: ...the impact of the COVID-19 pandemic on influenza vaccination uptake among pregnant women. Please consider ‘uptake’ instead of or along with ‘acceptance’.

Line 290-292: Reference 65 is contradictory to the findings of the present study. Should be stated as such.

Reviewer 2 Report

 Abstract 

*** How was acceptance of influenza vaccine not influenced by the COVID pandemic if you look at the increased odds of 6.11 for covid vaccination compared to the odds in other factors Kindly revise lines 24-26 so it conforms to the results presented.

Introduction

Merged lines 30-31 with lines 41 -43. Since initial sentence in lines 41-43 will be moved up, add new sentence where you provide the burden of covid in Korea, the burden of influenza in Korea and how covid has impacted on the burden of influenza especially among pregnant women.

Methods

Well written, just add to the sub section on study design and participants how sample size estimation was made.

Section 2.2.4 which is “Trust in healthcare providers” in line 103 needs more definitions. Consider providing more details on what the two items used to assess trust because some questions because of this could trust have been due to how well awareness campaign during the covid was performed in addition to how satisfied the participants were with care provided in the covid era compared to the pre covid era? 

Results

Sub-section 3.1, if of the total participants, 51.0% were vaccinated against influenza during pregnancy and most 134 women had a history of influenza vaccination (n=327, 93.2%), how then was it important to access the acceptance of the influenza vaccine? With this, I recommend for there to be a table 1 which should be only description of the study participants and for current table 1 to be Table 2. 

Very good analysis but again, with regards to the objective of this paper, I suggest for the authors to work with a statistician to ensure the analysis aligns with the objective.  Only section 3.2 speaks to the objective. The other sections raise confusion on if the authors just wanted to find out facilitators of influenza vaccine uptake irrespective of the covid pandemic. The authors should have a look at this paper - https://www.ncbi.nlm.nih.gov/pmc/articles/PMC8048709/  to guide the revision. For example, the acceptability in Current table 1 first row after all considerations to re-analyse, can be revised to “Acceptance of influenza vaccine during pregnancy due to covid”. Across all other tables, it should be clear that the analysis was based on acceptance of influenza vaccine due to covid. If this major revision is not possible due to data, then consider revising your objective in lines 62-66 to be solely “factors or facilitations of influenza vaccine uptake among pregnant women because this is what the current result section seems to indicate. 

Lastly, 

Once the authors clarify and address the above comments for the result sections, I will read the discussion. Thanks

Reviewer 3 Report

Impact of the COVID-19 Pandemic on Influenza Vaccination 2 and Associated Factors among Pregnant Women: A Cross-Sectional Study in Korea

My suggestions for this study are given below.

First of all, it is important to research information about the covid 19 vaccine and influenza vaccine in pregnant women. Although the number of people participating in the survey is low, it can be a guide in terms of taking a picture of the situation.

In this study, it was reported that pregnant women under 12 weeks were excluded from the study in the material method section. However, the following analysis is included in Table 1. It is recommended to rearrange the table.

Method: (Sentence 74. 75). Individuals under 12 weeks of gestation or who were giving birth to their first child were excluded  from the analysis.)

Table 1. First trimester (1-12 week)

14     7.0), 7 (7.3), 7 (6.7)

1.       (Sentence 265-266)In the present study, trust in healthcare providers was significantly higher in pregnant women vaccinated against influenza than in unvaccinated pregnant women.

It was concluded that the confidence in health workers was higher in pregnant women who were vaccinated. It would be appropriate to add to the analysis whether this result is related to the increase in the educational status of the pregnant woman.

2.       Suggestions to remove the following article as it is not directly related to this study.

Sentence 292-293.

Another study conducted among adults aged 50–96 years in Can-aa found that those who received the influenza vaccine are significantly more likely to report willingness to receive the COVID-19 vaccine (OR 14.3; 95% CI 12.5, 16.2) [66].

Round 2

Reviewer 2 Report

I am satisfied with the author's responses.